# Large-Scale Fabrication of Graded Convex Structure for Superhydrophobic Coating Inspired by Nature

**DOI:** 10.3390/ma15062179

**Published:** 2022-03-16

**Authors:** Yu Wang, Jin-Tian Huang

**Affiliations:** Key Laboratory of Resource Fiber Utilization of Psammophytes, Inner Mongolia Agricultural University, Hohhot 010018, China; yuw@emails.imau.edu.cn

**Keywords:** superhydrophobic, coating, durability, self-cleaning

## Abstract

The addition of toxic substances and poor durability severely limit the market applications of superhydrophobic coatings in the oil–water-separation industry, anti-icing, and self-cleaning surfaces. In order to solve the above problems, a stable, strong, fluorine-free superhydrophobic coating was prepared according to natural inspiration. In this study, polydivinylbenzene (PDVB) was produced by the hydrothermal method, and micro-nanoparticle clusters composed of PDVB particles of different sizes were prepared by controlling the ratio of raw materials, which was then attached to the substrate surface by a simple spraying technique. A rough coating with a lotus-leaf-like layered protruding structure was constructed by depositing particle clusters of different sizes. In the end, the prepared coating showed attractive superhydrophobicity, with a maximum contact angle (CA) that reached up to 160°. In addition, the coating had long-lasting superhydrophobic properties in various environments, such as common liquid and acidic and alkaline solutions. Moreover, in the oil–water-separation process, the superhydrophobic filter paper was still able to obtain a separation efficiency of more than 85% after being used 50 times, and it maintained a contact angle of >150°. At the same time, the coating had excellent dye resistance and self-cleaning performance.

## 1. Introduction

The superhydrophobic phenomenon usually refers to the phenomenon of water droplets falling onto the surface of a sample with contact angles of 150° or more and rolling angles of less than 10°. In the natural environment, there are numerous superhydrophobic phenomena, such as the leaves of plants [1,2,3,4], the wings of butterflies, and the feet of insects [5], among which the surfaces of lotus leaves are the most widely known. According to the surface-structure characteristics of the lotus-leaf effect, other researchers have summarized two necessary conditions to prepare a superhydrophobic surface [6,7,8]: a micro-nano double-rough structure on the surface of the material, which is modified with low-surface-energy substances. Based on the above principles, the biomimetic design of the superhydrophobic surface inspired by superhydrophobic structures in nature has considerable value and is expected to be popularized and widely used in self-cleaning [9,10,11,12], antifouling [13,14] and water–oil separation [15,16,17] and other industries.

In previous studies, typical methods for preparing superhydrophobic surfaces have included the etching method [18,19], electrospinning [20,21], the sol-gel method [22,23], and physical- and chemical-deposition methods [24,25]. Chen et al. [26] combined submicron silica particles with different diameters in the form of covalent bonds, and then modified them with polydimethylsiloxane. Through the impregnation method, a durable and chemically stable superhydrophobic fabric was prepared, which had excellent oil–water-separation performance. Han [27] thermally induced the formation of polydopamine (Pdop) particles on a copper mesh or sponge (MF), and after modification with octadecylamine (ODA), a metal mesh with superhydrophobic/superlipophilic properties was obtained. However, complicated operating procedures and expensive equipment have greatly hindered the introduction of products into the market. To this end, some improved technologies have been developed in order to produce superhydrophobic surfaces, among which spraying is a means to achieve large-scale production. Huang et al. [28] sprayed an ethanol suspension of nanocellulose filaments beforehand. The surface of the wood treated with the adhesive was modified by chemical-vapor precipitation to obtain a superhydrophobic coating of nano-cellulose filaments. However, the micro-nano structure and the durability of low-surface-energy substances are key factors that determine whether the spraying method can be widely used to prepare superhydrophobic coatings and market them.

In this study, a uniform superhydrophobic coating that mimics the surface of the lotus leaves with a micro-nano structure was constructed through effective spraying. As demonstrated in the mechanism diagram 1, polydivinylbenzene (PDVB) of different particle sizes were prepared by a hydrothermal reaction combined with ultrasonic treatment, then mixed in different proportions and sprayed onto the surface of the substrate to which the adhesive had been pre-attached. The surface imitated the micro-nano convex structure of the lotus-leaf surface, which not only significantly improved the surface roughness of the substrate, but also created a large amount of air gaps between the water droplets and the coating surface, resulting in superhydrophobic properties. Moreover, the micro-nano structure composed of countless small particles has good abrasion resistance. When the outermost nano-sized balls are worn out, new nano-sized balls are exposed, so the superhydrophobic properties can be preserved. The method is simple, economical, and scalable, and the superhydrophobic coating shows considerable application prospects for water–oil separation.

## 2. Experimental Section

### 2.1. Materials Synthesis

All solvents were analytically pure reagents. Tetrahydrofuran, methylene-blue, Sudan-4, n-hexane, filter paper, divinylbenzene (DVB), azobisisobutyronitrile (AIBN) and ethanol were all purchased from Tianjin Sinopharm Chemistry Co., Ltd. (Tianjin, China). Tetrahydrofuran and commercial adhesive (HSM) (the main ingredient was acrylate) were obtained from McLean Chemical Reagent Co., Ltd. (Beijing, China) All of the above chemical reagents were analytical grade.

#### 2.1.1. Fabrication of Polydivinylbenzene Nanoparticles

DVB (2.0 g) of was dissolved in a 20 mL mixed solution of tetrahydrofuran and ethanol with a volume ratio of 10:1. AIBN (0.05 g) was added to the above mixture, stirred evenly and transferred to a small stainless-steel reactor with PTFE lining. Before the reaction, the air in the autoclave was replaced with high-purity nitrogen. Then, it was sealed and placed in an electric drying oven (101-2S, Shanghai Yiheng Scientific Instrument Co., Ltd., Shanghai, China) at 100 °C for 24 h, followed by suction filtration and vacuum drying. White block polydivinylbenzene (PDVB) solid material was obtained. PDVB-NPs of different particle sizes were prepared by controlling the mass ratios of DVB to AIBN (20, 2, and 0.2, respectively). The corresponding compounds were defined as P-y, where y is the mass ratio of DVB to AIBN used in the above reaction process.

#### 2.1.2. Fabrication of Superhydrophobic Coating

A certain quality of PDVB was added to the ethanol solution, and then ultrasonically broken for 1 h to prepare suspensions with different particle diameters. First, the surface of the substrate was treated with commercial adhesive. After 10 min, a commercial spray gun was used to spray the ethanol dispersion of PDVB-NPs onto the surface of the adhesive, followed by drying at ambient temperature for 20 min to completely volatilize the ethanol, thereby successfully obtaining a superhydrophobic coating. The preparation process is shown in Figure 1.

### 2.2. Measurements and Characterizations

A scanning electron microscope (Hitachi S4800, Shanghai, China) was used to study the morphology of coating surfaces; an atomic-force microscope (BRUKER Dimension Icon, Karlsruhe, Germany) was used to measure the surface roughness. The contact angles (CAs) and sliding angles (SAs) were measured using the appropriate measuring equipment (SL200, Shanghai Fangrui Instrument Co., Ltd., Shanghai, China). The size distribution of the PDVB-NPs was measured through a dynamic-light-scattering instrument (DynaPro NanoStar, Santa Barbara, CA, USA). The chemical structure of PDVB was characterized using a Fourier Infrared (FT-IR) Spectrometer (Nicolet Magna-IR 750, Madison, WI, USA). The spectra wavenumbers for the measurements were in the range of 4000−500 cm^−1^.

### 2.3. Stability Testing of the Superhydrophobic Coatings

Multiple washing and sandpaper-rubbing steps were continuously used to evaluate the stability of the superhydrophobic coating. The washing step was performed by immersing the superhydrophobic surface in deionized water and stirring at 500 rpm and 40 °C for 60 min. Then, the superhydrophobic surface was removed from the solution and dried at 100 °C for 0.2 h. The whole process was defined as the washing process. In order to carry out the abrasion test, the superhydrophobic filter paper was placed on 1000-grit sandpaper, and a weight load of 80 g was maintained. For one wear cycle, the slide was slowly moved 8 cm in the horizontal direction.

The corrosion-resistance test of the coating was carried out by immersion in several common solutions, such as strong acid (pH = 1), coffee, milk, green tea, and an alkali solution (pH = 14) for 12 h. After rinsing and immersing the superhydrophobic coating with deionized water to remove excess solvent, the sample was dried at 100 °C for 2 h and the CA was measured immediately.

The anti-ultraviolet performance of the superhydrophobic coating was evaluated by continuously exposing the sample to a UV desktop lamp for 96 h; the distance between the coating and the UV-light source was 15 cm, and the CA was measured every 12 h.

### 2.4. Oil−Water Separation Experiment

The oil–water selectivity test was carried out using self-made equipment. The cellulose filter paper with the superhydrophobic coating was placed on a conical flask. The superhydrophobic filter paper (9 cm) was considered as the filter in this device. During the test, n-hexane was used as the test oil. A volume of 25 mL of Sudan-4-coated oil was mixed with 25 mL of methylene-blue (Tianjin, China)-coated water to form an oil–water mixture, then poured into a glass funnel. Due to gravity, the oil sank in the collection container via the superhydrophobic filter paper. In order to study the reusable performance of the superhydrophobic filter paper, the separation process was repeated 50 times. After every cycle, the filter paper was washed with ethanol and dried before the next separation. According to the weight loss of the n-hexane before and after the separation, the separation efficiency was calculated.

## 3. Results and Discussion

### 3.1. Chemical Analysis, Morphology and Wettability

Figure 1 shows the infrared spectrum of the PDVB nanoparticles. The peak at 3054 cm^−1^ in Figure 1 corresponds to the stretching vibration of the C-H bond on the benzene ring, which is the characteristic peak of the benzene ring. The peaks at 1509 cm^−1^, 1486 cm^−1^, and 1451 cm^−1^ correspond to the stretching vibration of the benzene-ring skeleton. The peak at 3018 cm^−1^ corresponds to the stretching vibration of the C-H bond of vinyl, which is the characteristic peak of vinyl. The peaks at 2929 cm^−1^ and 2869 cm^−1^ correspond to the antisymmetric and symmetric stretching vibrations of the C-H bond of the methylene group. The peak at 1605 cm^−1^ corresponds to the stretching vibration of the vinyl C=C bond. This is the characteristic peak of methylene [29]. The existence of this series of peaks indicates that the polydivinylbenzene material was successfully synthesized by the hydrothermal method.

Figure 2 reveals the SEM images and size distribution of the P_40_, P_4_, and P_0.4_ PDVB-NP_S_. Obviously, the combined solvothermal reaction and ultrasonic disintegration did indeed produce nanoparticles (Figure 2a,c,e). Figure 2b,d,f depict the size distribution of the PDVB-NPs, in which two main peaks exist: the smaller peak is less than 100 nm, while the larger peak is greater than 500 nm. In other words, the larger peak indicates the PDVB-NPs cluster, and the smaller peak may indicate the size distribution of PDVB-NPs. Based on the smaller peak, we obtained average particle sizes of 81.5, 40.3, and 12.7 nm corresponding to P_40_, P_4_, and P_0.4_, respectively. The peaks of the PDVB-NPs clusters at 1200, 760, and 900 nm correspond to P_40_, P_4_, and P_0.4_, respectively. It is clear that the combination of solvothermal treatment and sonication does produce nanoparticles and micro-scale clusters, which is the key to the superhydrophobicity of the coating.

For the purpose of determining the effect of different mass ratios of PDVB-NP_S_ on the surface morphology of the superhydrophobic PDVB coatings, the morphologies of the filter paper coated with adhesive and the P_40_-coated filter paper (P_40_ particle content was 3 wt%) were studied. Filter paper coated with P_40_–P_4_ (based on the total content of 3 wt%, the mass ratio of P_40_ and P_4_ was 1:1) and filter paper coated with P_40_–P_4_–P_0.4_ (The mass ratio of P_40_, P_4_ and P_0.4_ particles was 1:1:1, based on the total content of 3 wt%) was studied through SEM (Figure 3). Obviously, the filter paper modified by PDVB with three different particle sizes showed the highest superhydrophobicity due to its special micro-nano structure.

Because surface roughness is the primary factor in constructing a superhydrophobic surface, the surface roughness of P_40_–P_4_–P_0.4_ superhydrophobic coating was further characterized by AFM. In Figure 4a,b, the surface of the adhesive-coated filter paper is very smooth, with a root-mean-square roughness (Rq) of 1.31 nm. The Rq value is usually used to quantitatively evaluate roughness; the higher the Rq value, the greater the roughness. Unlike the adhesive-coated filter paper, the P_40_–P_4_–P_0.4_ adhesive-coated filter paper that was made from PDVB-NP_S_ with different particle diameters shows a rougher profile and the shaped protrusion structure varies greatly in height (Figure 4c,d). The calculated Rq value is 65.2 nm. It is believed that the mountain-shaped convex structure traps the air in the grooves between the peaks, thereby forming a stable air layer. As the contact area between the water droplets and the coating surface is reduced, a hydrophobic surface with low adhesion is produced.

In this paper, CA is used as the main indicator to measure the hydrophobic performance of the coating. Figure 5a exhibits the relation between the particle content of P_40_, P_4_, and P_0.4_ and the CA of the superhydrophobic surfaces. When the content of PDVB-NP_S_ was different, the CA had a significant change from 95° to 152°. The rapid increase in the CA on the superhydrophobic coating was due to the increase in surface roughness. It is worth noting that when the content of PDVB-NP_S_ was 3 wt%, the optimal hydrophobicity reached 152°. When the content of PDVB-NP_S_ was less than or greater than 3 wt%, the CA was less than 152°, indicating that PDVB-NP_S_ had excessively accumulated on the surface. The content ratio of the three PDVB-NP_S_ based on 3 wt% total content was studied in order to maximize the CA. Two kinds of PDVB-NP_S_ were selected to prepare functionalized coatings, and the mass ratios were 1:3, 2:2, and 3:1, respectively. Under different combinations, when the mass ratio of the P_40_–P_0.4_ coating was 1:3, the maximum CA was 156.2° (Figure 5b). This should be due to the construction of an effective hierarchical protruding structure on the surface and the proper collaboration of clusters of different sizes. After the three kinds of particles were blended with various mass ratios, the CA of the functionalized coating was further improved. When the three PDVB-NP_S_ were mixed in different mass ratios, the minimum CA was 154° and the maximum CA was 160.1° (Figure 5c). The improved CA and SEM results were in good agreement, indicating that the composite of PDVB particles of various sizes had a key effect on the wettability of the coating. By adjusting the mass ratio of the PDVB-NP_S_ with different diameters, the best superhydrophobicity was obtained, and the lotus-leaf-like micro-nano multi-level protruding structure was successfully built on the surface.

In order to test the waterproof capability in actual application, the water droplets are almost spherical on the P_40_–P_4_–P_0.4_ superhydrophobic coating (Figure 6a). When the superhydrophobic coating was immersed in water, bubbles trapped by the micro-nano rice structure appeared on the surface of the coating. Attractively, because an air layer was trapped on the coating, an obvious silver-mirror phenomenon appears (Figure 6b). As the contact area between the coating and the water droplets was reduced, the air layer effectively blocked the coating after wetting. Even if high-speed water spray is sprayed onto the surface of the coating, the water can easily bounce off the coating (Figure 6c), which further proved the strong superhydrophobic properties of the coating. According to the Cassie–Bacchus law [30,31], this excellent non-wettability is due to the low adhesion of water which was caused by the air pockets captured by the micro-nano structure, which made it easy for water droplets to fall onto the superhydrophobic surface. Figure 6d shows the water-drop-bounce test. A 30 µL water droplet made a free fall motion from a height of 40 mm from the coating. After dropping onto the superhydrophobic coating, it quickly bounced, indicating that the superhydrophobic coating had extremely low water adhesion and impact resistance. It further proved the analysis results of the sputtering test of the water column on the superhydrophobic surface (Figure 6c). The impact resistance comes from the buffering and dispersing effect of the micro-nano convex structure on the impact force, and the low water adhesion is conducive to the efficient self-cleaning of the coating.

### 3.2. Durability and Chemical Stability of the Superhydrophobic Coating

When applied in practice, the superhydrophobic coating is usually exposed to a relatively complex and difficult environment. With the accumulation of mechanical damage and chemical corrosion, it may lose its superhydrophobic properties. A series of methods were implemented in order to measure the durability of the P_40_–P_4_–P_0.4_ superhydrophobic coating, such as the washing test, sandpaper friction and ultraviolet light irradiation. After 100 sandpaper-rubbing cycles, the superhydrophobic filter paper coated with P_40_–P_4_–P_0.4_ still showed excellent superhydrophobic properties, with a CA of 146.3° and satisfactory water resistance. Encouragingly, in severely worn areas, the water droplets were still nearly spherical (Figure 7a). These results show that the superhydrophobic coating is durable and has excellent abrasion resistance. The durability of the superhydrophobic coating was evaluated by continuous UV-light irradiation, and it was found that the CA was 153.4° after 96 h of continuous UV-light irradiation, indicating that the P_40_–P_4_–P_0.4_ superhydrophobic coating has excellent UV-aging resistance (Figure 7b). The washing durability was also tested by recording the CA fluctuation of the superhydrophobic coating of the P_40_–P_4_–P_0.4_ coating in different washing cycles (Figure 7c). As a result, the CA decreased slowly with the increase in number of washes times and was maintained at 145.2° after 40 washes, which indicates that the P_40_–P_4_–P_0.4_ superhydrophobic coating has excellent washing durability, which is due to the excellent stability of PDVB with a crosslinked structure to various extreme conditions.

In different surroundings (for instance, common liquids), the potential deterioration of the superhydrophobic coating is due to the weakening of the bonding strength between the superhydrophobic surface and the substrate. The corrosion resistance of the P_40_–P_4_–P_0.4_ superhydrophobic coating was evaluated in various solutions. The prepared superhydrophobic coating was immersed in strong acid, milk, coffee, green tea, and strong alkali solutions for 24 h, followed by measurement. As shown in Figure 7d, after immersion in various solutions for 24 h, the CA decreased slightly and remained above 150°.

It can be concluded that the prepared P_40_–P_4_–P_0.4_ superhydrophobic coatings have excellent stability and chemical resistance. The SEM images of the P_40_–P_4_–P_0.4_ coating before and after 50 washing cycles (Figure 8a,c) and 96 h UV irradiation (Figure 8b,d) can be observed. Comparing Figure 8a and Figure 8c, the surface roughness of the superhydrophobic coating is reduced, but the principal structure of the micro-nano protrusions still exists. After prolonged UV irradiation, the coating surface in Figure 8b,d showed little change, indicating the durable water repellency of the coating.

### 3.3. Self-Cleaning Capability of the Superhydrophobic Coating

In terms of the self-cleaning-capability evaluation, both pure-glass slides and superhydrophobic glass slides coated with P_40_–P_4_–P_0.4_ were contaminated with sawdust and then tested. The glass slides were tilted at the same angle so that water droplets fell onto the surface of the glass slide. The pure-glass slide quickly became heavily contaminated. On the contrary, all the powder on the superhydrophobic glass slides that were coated with P_40_–P_4_–P_0.4_ was immediately removed by rolling water droplets, so the clean glass-slide surface maintained its original state (Figure 9a,b). Therefore, it was confirmed that the P_40_–P_4_–P_0.4_ coating possesses good self-cleaning ability.

### 3.4. Oil−Water Separation Capability of the Superhydrophobic Coating

Due to the excellent superhydrophobic capability of P_40_–P_4_–P_0.4_ coating, it is not only hydrophobic but also lipophilic. Therefore, P_40_–P_4_–P_0.4_-treated paper has high potential in oil–water-separation applications. The water and oil wettability of the P_40_–P_4_–P_0.4_-coated superhydrophobic filter paper initially proved its potential (Figure 10a,b). As can be seen, the surface of the filter paper has superhydrophobic and superlipophilic properties, and the water droplets and oil droplets on the filter paper were immediately absorbed, but only the oil droplets were absorbed by the superhydrophobic filter paper coated with P_40_–P_4_–P_0.4,_ leaving the water droplets remaining as a spheres on the surface. In addition, both the filter paper and the superhydrophobic filter paper coated with P_40_–P_4_–P_0.4_ were immersed in a glass beaker containing n-hexane that was dyed with Sudan red and water dyed with methylene blue in order to study the oil–water-separation ability. Obviously, the n-hexane was completely absorbed by the superhydrophobic filter paper coated with P_40_–P_4_–P_0.4_, and there was no apparent oil remaining in the collection container (Figure 10c,d), proving that the superhydrophobic filter paper coated with P_40_–P_4_–P_0.4_ showed excellent hydrophobicity and lipophilicity. The oil–water mixture was poured into a solvent filter equipped with a P_40_–P_4_–P_0.4_-coated superhydrophobic cellulose filter paper (Figure 10e–g). The results show that only the red oil reached the flask through the superhydrophobic filter coated with P_40_–P_4_–P_0.4_, but the blue-dyed water was thoroughly prohibited and stayed in the funnel. After separation, there was no obvious blue water remaining in the collection container and no red oil in the funnel, which confirmed that the superhydrophobic filter coated with P_40_–P_4_–P_0.4_ had excellent oil–water-separation performance.

In Figure 11a, after 50 cycles of separation, the separation capability declined but maintained above 85%, indicating an excellent separation capability of the prepared P_40_–P_4_–P_0.4_ superhydrophobic filter paper in oil–water separation applications. Moreover, the CA of the P_40_–P_4_–P_0.4_-coated superhydrophobic filter paper after multiple separation cycles of oil–water mixtures was tested in order to prove the recyclability and reusability of oil–water separation. The CA of the superhydrophobic filter paper coated with P_40_–P_4_–P_0.4_ remained at 145°. The above separation cycle (Figure 11b) confirms the strong oil–water-separation capability of the superhydrophobic filter paper coated with P_40_–P_4_–P_0.4_.

## 4. Conclusions

In summary, a highly durable P_40_–P_4_–P_0.4_-coated superhydrophobic coating was prepared via a simple hydrothermal and spraying strategy. The PDVB clusters composed of three kinds of nanoparticles with different particle diameters were combined together in order to successfully construct a superhydrophobic coating with a uniformly layered and prominent structure. The P_40_–P_4_–P_0.4_ coating showed good superhydrophobicity; the highest CA was about 160°. Besides, the large number of micro-nano-level protrusions composed of PDVB particles provided a steady stream of superhydrophobic structures, resulting in outstanding durability to endure even man-made damage and long-term ultraviolet-light exposure. At the same time, the prepared P_40_–P_4_–P_0.4_ superhydrophobic coating had excellent self-cleaning ability. After 50 oil–water-separation cycles, the efficiency still remained above 85%, and the CA was greater than 145°. From the perspectives of wettability, durability and potential large-scale production feasibility, the strategy designed and the P_40_–P_4_–P_0.4_ superhydrophobic coatings prepared in this study have a wide range of application prospects in the fields of self-cleaning and oil–water separation. The hydrothermal method combined with mechanical treatment to prepare different particle diameters of polydivinylbenzene (PDVB) clusters with different particles is simple and easy to implement in large-scale fabrication. Moreover, spraying is a means of capably of achieving the fast and large-area preparation of superhydrophobic surfaces. The excellent durability and low cost of the technology increase their marketability.

## Data Availability

The data that support the findings of this study are available from the corresponding author upon reasonable request.

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
