# Peer review of "Large-Scale Fabrication of Graded Convex Structure for Superhydrophobic Coating Inspired by Nature"

_materials, 2022, doi:10.3390/ma15062179_

Round 1

Reviewer 1 Report

The authors report on fabrication of superhydrophobic coating based on a mixture of polydivinylbenzene (PDVB) micro- and nanoparticles by spraying technique. The obtained hierarchical surface relief due to a fine balance of particles of different sizes within particle clusters formed on a substrate provide a number of attracting properties, such as high contact angles, self-cleaning ability, oil-water separation together with high resistance to external damaging effects. The material obtained is very promising for practical applications. My main concern is that the article gives the impression of a set of tests, the results of which are described, but not explained. What is the role of the adhesive used? What is the height of the particle layer on the surface? What is the key factor defining the superhydrophobicity? What is the accuracy in contact angle measurements? Whether the results presented (for instance, fig. 5) are mean values? What are the sliding angles? The self-cleaning and oil-water separation tests are purely qualitative. I would recommend the authors the major revision before publication. Also please check the manuscript for misprints, there are a lot of typos.

Author Response

Dear reviewer:

Thank you for your kind attention to consider our manuscript entitled “ Large-Scale Fabrication of Graded Convex Structure for Superhydrophobic Coating Inspired by Nature”. (No: materials-1578845). The following are the responses to the your comments.

 (1) My main concern is that the article gives the impression of a set of tests, the results of which are described, but not explained.

Answer: In this study, the highlight of the experimental design is that for the first time, PDVB particle clusters with lotus-shaped micro-nano protrusions were prepared by a combination of hydrothermal reaction and mechanical treatment, and were used for the preparation of superhydrophobic coatings. As expected, the material has encouraging properties and is expected to achieve scale preparation.  However, the formation mechanism of micro-nano-scale PDVB particle clusters and the formation mechanism of superhydrophobicity are not sufficiently analyzed. To this end, the authors provide additional clarifications and analyses in the relevant sections of the manuscript.

(2) What is the role of the adhesive used?

Answer:  The role of the adhesive in this study is to connect the treated substrate surface with the PDVB particle clusters.

(3) What is the height of the particle layer on the surface?

Answer: In this experiment, the thickness of the superhydrophobic coating depends on the concentration of PDVB ethanol suspension, the distance between the airbrush and the treated surface, and the spraying time. The thickness of the coating with the best superhydrophobicity (P40-P4-P0.4 superhydrophobic coating) is between 2.3-3.6μm.

(4) What is the key factor defining the superhydrophobicity?

Answer: A superhydrophobic surface is defined as a surface with a water contact angle (CA) greater than 150° and a sliding angle (SA) less than 10°.

(5) What is the accuracy in contact angle measurements?

Answer: In this study, a contact angle measurement instrument (SL200, China) was used to detect the contact angles and sliding angles were ed with. According to the technical parameters and test results provided by the manufacturer. The accuracy in contact angle measurements is 0.1°.

(6) Whether the results presented (for instance, fig. 5) are mean values?

Answer: In the study, all the results presented are mean values.

(7) What are the sliding angles?

Answer: In the study, droplet method is used to test the sliding angles. Due to the high superhydrophobicity of the PDVB-treated coating, when the liquid dropped from the needle tube to the tested surface, it was difficult to keep on the surface but rolled down. It is difficult for the experimenter to obtain a more accurate value when testing the contact angle. Therefore, only some sliding angles with strong reliability are provided.

(8) Also please check the manuscript for misprints, there are a lot of typos.

Answer: The authors have carefully checked the manuscript for writing errors and corrected them.

Reviewer 2 Report

This manuscript reports on manufacturing superhydrophobic coatings, whose microstructure is reminiscent of the lotus leaf. The coating with the maximal hydrophobicity reaches the contact angle of 160 degrees. The analysis shows excellent self-cleaning ability and outstanding durability even after a series of damages. The authors believe that the simplicity of their spraying coating technique is potentially attractive for large-scale production for superhydrophobic coating.

The procedures are well described, the analysis solid, and the results very appealing. I support the publication of the manuscript.

I only have a few minor remarks:

  1. Do contact angles on these coatings depend on the droplet size? In other words, is the line tension measurable and relevant for this length scale?
  2. In Figs. 5b and 5c, the differences between the bars are not well visible. Consider plotting a narrower range of contact angles.
  3. The acronyms PDVB and CA, introduced in the abstract, have to be introduced independently in the main text at first appearance.

Author Response

Dear reviewer:

Thank you for your kind attention to consider our manuscript entitled “ Large-Scale Fabrication of Graded Convex Structure for Superhydrophobic Coating Inspired by Nature”. (No: materials-1578845). The following are the responses to the your comments.

 (1) Do contact angles on these coatings depend on the droplet size? In other words, is the line tension measurable and relevant for this length scale?

Answer: First, the magnitude of the contact angle is related to the volume of the droplet. The droplet volume should be controlled between 0.5 and 10 uL, to avoid the droplet deformation due to excessive weight or the possibility of rolling off the hydrophobic surface; after the liquid is deposited on the surface of the sample, it is generally considered that the standing time is 20 s It has smaller evaporation and better stability [1]. The droplet volumes used in this study were all 0.5 uL. Secondly, Thomas Young believes that when a liquid drops on a solid surface, it will be subject to the combined action of its own gravity, internal force and surface tension of gas, liquid and solid three-phase, and the contact angle can be measured when it reaches a stable state, and proposed Young's equation:

                                                 γsv- γsl= γlvcosθe

where γsv is the solid-gas surface tension; γsl is the solid-liquid surface tension; γlv is the liquid-gas surface tension; θe is the intrinsic contact angle.

  Therefore, the tension of a liquid is closely related to its contact angle.

(2) In Figs. 5b and 5c, the differences between the bars are not well visible. Consider plotting a narrower range of contact angles.

Answer: In Figures 5a and 5c, the differences between the bars are indeed not well visible, but using a narrower contact angle range compresses the space for text labels. In order to clearly represent the experimental data while taking into account the readability of the manuscript, the authors filled in the contact angle values in Figures 5a and 5c in the form of supplementary data in a table named Table S1.

(3) The acronyms PDVB and CA, introduced in the abstract, have to be introduced independently in the main text at first appearance.

Answer: The authors have supplemented their acronyms PDVB and CA where polydivinylbenzene and contact angle are first mentioned in the abstract.

[1] Kubiak K, Wilson M, Mathia T, et al.Dynamics of contact line motion during the wetting of rough surfaces and correlation with topographical surface parameters [ J].Scanning, 2011, 33(5): 370 377.

Reviewer 3 Report

Decision: minor revision

The work presents the preparation of superhydrophobic coating via a simple hydrothermal and spraying strategy. The described method is simple, economical, and scalable, for obtaining of superhydrophobic coating. A few corrections need to be made before publication:

Line 67 - correct: water;

Line 77 - DVB - is not powder;

Line 79 - What was replaced by nitrogen?

Line 80 - "a constant temperature" should be removed.

Also, in my opinion, the journal "Coatings" (MDPI) is more suitable for publication of this manuscript. 

Author Response

Dear reviewer:

Thank you for your kind attention to consider our manuscript entitled “ Large-Scale Fabrication of Graded Convex Structure for Superhydrophobic Coating Inspired by Nature”. (No: materials-1578845). The following are the responses to the your comments.

 (1) Line 67 - correct: water.

Answer: The spelling error "wate-r oil" in line 67 has been changed to "water-oil", the modification is marked in red font.

(2) Line 77 - DVB - is not powder;

Answer: Thank you very much for your careful review of our manuscript, DVB is indeed not a powder but a liquid. This misstatement has been corrected throughout the text. The modification is marked in red font.

(3) Line 79 - What was replaced by nitrogen?

Answer: The synthesis of PDVB is carried out in a high temperature hydrothermal reactor, which needs to avoid contact with air. Therefore, it is necessary to replace the air in the reactor with high-purity nitrogen before the reaction is carried out. The authors have stated in the corresponding section of the manuscript, shown in red font.

(4) Line 80 - "a constant temperature" should be removed.

Answer: – the redundant description "a constant temperature" in Line 80 has be removed.

Reviewer 4 Report

This paper deals with the preparation and characterization of a  fluorine-free superhydrophobic coating.

The paper can be published after minor revisions.

-line 67:  please correct the spelling error wate-r oil

- In figure 3 it is difficult to read the letters in the pictures.

- In figure 8 it is better  to show  the original surface for comparison and to maintain the same magnification (1 micron mark)

Author Response

Dear reviewer:

Thank you for your kind attention to consider our manuscript entitled “ Large-Scale Fabrication of Graded Convex Structure for Superhydrophobic Coating Inspired by Nature”. (No: materials-1578845). The following are the responses to the your comments.

 (1) -line 67:  please correct the spelling error wate-r oil.

Answer: The spelling error "wate-r oil" in line 67 has been changed to "water-oil", the modification is marked in red font.

(2) - In figure 3 it is difficult to read the letters in the pictures.

Answer: Since the original SEM image has not been modified, it is difficult to see the text and ruler in Figure 3. After careful revision, the authors have made changes to all the original figures to ensure the readability of the manuscript.

(3) - In figure 8 it is better  to show  the original surface for comparison and to maintain the same magnification (1 micron mark)

Answer: Indeed, Figure 8 miss the surface topography of the coating before abrasion and UV exposure as a comparison. The authors have supplemented them and ensured the same magnification.